# Preparation and Phytotoxicity Evaluation of Cellulose Acetate Nanoparticles

**DOI:** 10.3390/polym14225022

**Published:** 2022-11-19

**Authors:** Regiane G. Lima, Maria Maranni, Leandro O. Araujo, Bruno Marinho Maciel, Thalita Canassa, Anderson R. L. Caires, Cícero Cena

**Affiliations:** Optics and Photonic Lab (SISFOTON-UFMS), Instituto de Física, UFMS-Universidade Federal de Mato Grosso do Sul, Campo Grande 79070-900, Brazil

**Keywords:** non-phytotoxic, cellulose acetate, polymeric nanoparticles, *Allium cepa*

## Abstract

The use of biocompatible and low-cost polymeric matrices to produce non-phytotoxic nanoparticles for delivery systems is a promising alternative for good practices in agriculture management and biotechnological applications. In this context, there is still a lack of studies devoted to producing low-cost polymeric nanoparticles that exhibit non-phytotoxic properties. Among the different polymeric matrices that can be used to produce low-cost nanoparticles, we can highlight the potential application of cellulose acetate, a natural biopolymer with biocompatible and biodegradable properties, which has already been used as fibers, membranes, and films in different agricultural and biotechnological applications. Here, we provided a simple and low-cost route to produce cellulose acetate nanoparticles (CA-NPs), by modified emulsification solvent evaporation technique, with a main diameter of around 200 nm and a spherical and smooth morphology for potential use as agrochemical nanocarriers. The non-phytotoxic properties of the produced cellulose acetate nanoparticles were proved by performing a plant toxic test by *Allium cepa* assay. The cytotoxicity and genotoxicity tests allowed us to evaluate the mitotic process, chromosomal abnormalities, inhibition/delay in root growth, and micronucleus induction. In summary, the results demonstrated that CA-NPs did not induce phytotoxic, cytotoxic, or genotoxic effects, and they did not promote changes in the root elongation, germination or in the mitotic, chromosomal aberration, and micronucleus indices. Consequently, the present findings indicated that CA-NPs can be potentially used as environmentally friendly nanoparticles.

## 1. Introduction

Cellulose is a natural organic polymer easily obtained by the extraction of plants, bacteria, and tunicates, and it is considered the most abundant natural biopolymer in the world, comprising 40–50% of the earth’s total biomass reserves. Cellulose stands out as a renewable, biodegradable, and non-toxic biopolymer [1], able to undergo acylation, etherification, oxidation, and deoxygenation reactions due to its hydroxyl groups [2].

Cellulose acetate (CA) is a cellulose-derived ester with biocompatible and biodegradable properties, with limited chemical modifications [3]. Cellulose-derived polymers are promising candidates for bio-based materials and replace the usual non-biodegradable polymeric materials, contributing to decreasing the levels of environmental pollution [4]. Among all cellulose derivatives, cellulose acetate has been recognized as an important cellulose derivative due to its mechanical strength, which facilitates its processing into films, membranes, and fibers from melts or solutions [4,5,6] for wide industrial and commercial applications [7]. The toxicity effect of cellulose acetate sheets in plants was first evaluated in 1952 by Karl Maramorosch [8]; since then, cellulose acetate-based materials have been explored for potential use in biological applications [9], for adsorbing material for water treatment [10], and as analytical tools [11], but the literature to evaluate the toxicity effects of cellulose acetate at the nanoscale remains sparse.

Recently, different cellulose acetate materials became promising solutions for agriculture-related applications. A cellulose-acetate-modified ionic liquid membrane was used as the adsorbent for the removal of “Pirimicarb”, an insecticide, from wastewater [5]. Hydrogel that was cellulose-acetate-derived and EDTAD were used as a substrate for NPK fertilizer in the soil and was able to reduce the leaching of fertilizers and improve the performance of eucalyptus seedlings [12]. Triolein-embedded cellulose acetate membranes reduced the plant bioavailability of polychlorinated biphenyls (PCBs) in soil [13].

Cellulose acetate (CA) is also considered a good candidate for biotechnology applications due to its low cost, hydrophilicity, non-toxicity, good processability, and biodegradable and renewable properties. The existence of groups like hydroxyl, carboxyl, and ether groups in its main backbone chain provides ionic features, making it an excellent nanoreactor for blending countless functional and catalytic nanoparticles on its surface [14]. Studies have been conducted to improve CA properties by incorporating different materials: bioactive nanofillers [15]; essential-oil nanocapsules with antimicrobial activity [16]; nanocomposites for biomedical packing and food safety [17]; and nanoparticles as drug-delivery-system applications [18].

Besides several nanotechnology advances with CA applications, studies on cellulose acetate nanoparticles (CA-NPs) and their phytotoxic properties are still sparse. Since some toxicity effects of these nanoparticles in plants are decisive in motivating new applications such as agrichemical nanocarriers [19], we have to explore the role played by CA-derived materials at the nanoscale. Therefore, in this study, CA-NPs were successfully produced by using a modified emulsification solvent evaporation technique o/w (oil/water), and phytotoxic tests were performed by using the *Allium cepa* assay. Cytotoxicity and genotoxicity tests allowed the evaluation of the typical toxic behavior of the NPs (such as in the mitotic process, chromosomal abnormalities, inhibition/delay in root growth, and micronucleus induction) [20,21].

## 2. Materials and Methods

### 2.1. Materials

The chemical reagents were purchased from commercial sources (Sigma-Aldrich, San Luis, Missouri, USA and Synth, Diadema-SP, Brazil) and used without further purification. Cellulose acetate (Mw~50,000) and Polyvinyl alcohol surfactant were purchased Sigma-Aldrich (Mw = 30,000–70,000). Dichloromethane, from Synth, was used for the organic solution.

### 2.2. Synthesis of Nanoparticles

Cellulose acetate nanoparticles were produced by using a modified emulsification solvent evaporation technique *o*/*w* (oil/water) [22]. Cellulose acetate raw material was added to a 2.2% (*w*/*v*) aqueous solution of polyvinyl alcohol (PVA). The method of adding the organic phase to the aqueous phase was used: adding in a single aliquot while stirring for 5 min at 1000 rpm and then applying Sonifer^®^ Branson SFX550 (Branson Ultrasonics, Danbury, Connecticut, USA) sonicate for 90 s at 30% A, with a pulsed vibration of 10” and a 0.5” tip. The suspension of the resulting NPs were stirred for 24 h at room temperature to completely evaporate the organic solvent. The NPs were centrifuged (15 min at 14,000 rpm) and washed three times with Millipore water. The obtained NP suspension was frozen in liquid nitrogen and lyophilized at −55 °C and 0.5 kPa. The powdered NPs were stored at −10 °C until use.

### 2.3. Hydrodynamic Particles Size and Zeta Potential Characterization

The hydrodynamic size distribution and polydispersity index (PDI) were measured by dynamic light scattering (DLS) using a light scattering instrument at a 90° scattering angle in a Malvern Zetasizer Nano ZS90 equipment (Malvern, WorcesTershire, UK). The refractive index was 1330, and the temperature was kept at 25 °C. These analyses were performed in triplicate at room temperature, each one comprising 15 runs of 10 s with a thermal stabilization period of 120 s. The nanoparticles were lyophilized in solution and dispersed in Mili-Q water. NP’s zeta potential (ξ) measurements were determined by laser Doppler electrophoresis technique. The zeta potential (ξ) measurements reflect the surface potential of the particles, checking the stability of the systems.

### 2.4. NPs’ Morphology Characterization

Scanning electron microscopy (SEM) of the CA-NPs was carried out in a JEOL JSM- 6380 LV microscope (JEOL, Akishima, Tokyo, Japan). The CA-NPs’ SEM images were obtained after NPs’ lyophilization and deposition on a metal stub; a conductive (Au) thin film was sprayed onto the sample surface by a sputtering technique. The morphological evaluations of the CA-NPs were performed by using SEM images with a 20,000- and 50,000-fold increase, and NPs’ diameter distribution was calculated based on the measurements of 100 particles randomly chosen by using the ImageJ software.

### 2.5. NPs’ Identification—Molecular Infrared Spectra

The molecular identification of CA-NPs was performed in a Perkin–Elmer (Perkin-Elmer, Waltham, Massachusetts, USA) infrared spectrometer, model spectrum 100, in the region of 4000 to 700 cm^−1^, with an attenuated total reflectance (ATR) accessory, by using 10 scans and a 4 cm^−1^ resolution. The lyophilized NPs and raw cellulose acetate material were investigated for comparison.

### 2.6. Allium Cepa Assay

The *Allium cepa* assay was performed by using pesticide-free seeds (Baia Periforme variety), purchased from the company Isla (Isla Sementes Ltda, Porto Alegre-RS, Brazil). For each group studied, 30 *Allium cepa* seeds were placed in Petri dishes with a layer of filter paper and were continuously exposed to an aqueous solution of CA-NPs, at concentrations of 12.5, 25, 50, and 100 μg/mL. Then, the seeds were placed in a germination and growth chamber, type B.O.D (Biochemical Oxygen Demand), with a temperature control of 25 °C, a humidity of 85 ± 5%, and lighting of 12 h/day for 4 days. A negative control group was also considered, in which the seeds were exposed to distilled water during germination. Thus, the experiment was divided into groups of CA-NP concentrations and the negative control. All tests were performed in triplicate, totaling 90 seeds per group.

The germination index (*GI*) was calculated after 96 h using Equation (1):
(1)$$GI=\frac{Total\;of\;seeds\;that\;germinated}{Total\;of\;seeds\;exposed\;to\;treatments\;}\times 100$$


The root elongation index (REI) was determined by calculating the arithmetic mean of the root sizes of each group, using a Digimess^®^ digital caliper after 96 h of seed exposure for germination. The roots of each treatment were collected and fixed in a Falcon tube together with 20 mL of Carnoy 3:1 (absolute ethanol/glacial acetic acid (*v*/*v*)). After 8 h, the fixative was removed and 20 mL of a new Carnoy fixative was added and stored at 4 °C. Subsequently, the roots were washed with distilled water, followed by acid hydrolysis with 1 M HCl at 60 °C for 10 min, and again they were washed repeatedly with distilled water. After a while, they were immersed in a solution of Schiff reagent, resting for 2 h. Subsequently, to prepare the slides, the meristematic regions of the *Allium cepa* roots were collected and sectioned with a scalpel and placed on a glass slide, and then a drop of 45% Acetic Carmine was added, covered with a coverslip, and carefully crushed. For each treatment, 5 slides were prepared and then observed under a Nikon optical microscope at 400× magnification, so that 1000 cells were counted per slide, totaling 5000 cells. Briefly, the *Allium cepa* test is represented in the schematic diagram in Figure 1.

The mitotic index (*MI*), chromosomal aberration index (*CAI*), and micronuclei index (*MNI*) was estimated as described by Equations (2)–(4):
(2)$$MI=\frac{Total\;of\;cells\;in\;division}{Total\;of\;cells\;observed}\times 100$$
(3)$$CAI=\frac{Total\;of\;cells\;with\;chromosomal\;abnormalities}{Total\;of\;cells\;observed}\times 100$$
(4)$$MNI=\frac{Total\;of\;micronucleus-bearing\;cells\;}{Total\;of\;cells\;observed}\times 100$$


## 3. Results and Discussion

The CA-NPs’ surface morphology was examined by SEM images, as shown in Figure 2A, showing nanoparticles with a regular spherical shape and a smooth surface. In Figure 2B, the CA-NPs’ diameter histogram shows a large distribution from 100 to 600 nm, with a mean diameter of around 216 ± 8.21 nm.

The particle size distribution determined by DLS based on intensity (Table 1) (D_Size-DLS_) and the Z average hydrodynamic size (D_Z-Average-DLS_) are consistent with the particle sizes observed from the SEM micrographs shown in Figure 2A and their respective particle distribution, shown in Figure 2B. The nanoparticles presented some polydispersity and may contain aggregated particles. The zeta potential of CA-NPs measured at neutral pH indicates negatively charged surfaces due to the presence of hydroxyl and carboxylic groups on their surfaces, with a relatively colloidal stability [23]. For all nanoparticles, the zeta potential values indicate a good stability of the nanoparticle systems due to effective electrostatic particle repulsion [15,24].

The cellulose acetate nanoparticles and cellulose acetate from Sigma-Aldrich FTIR spectra, shown in Figure 3, exhibit a high similarity. The CA’s characteristic bands are in the spectral range of 1745 cm^−1^ (elongation vibration of C=O groups), 1433.5 and 1370.5 cm^−1^ (C-H deformation vibration), and 1231 and 1044.5 cm^−1^ (elongation vibration of the C-O groups) [25,26,27]. At 1645 cm^−1^, it is possible to observe an H-O-H splitting band, associated with water absorption and widened in the nanoparticles’ spectra [6]. Only a small difference was identified for the 1512 cm^−1^ band, assigned to C-H groups’ stretching vibration [25], which was not observed for CA-NPs.

The effects of CA-NPs on the root elongation (REI) and germination indices (GI) are presented in Table 2. CA-NPs did not lead to a significant reduction in GI and REI, which indicates that CA-NPs do not induce phytotoxic effects. Although a relative reduction in the germination index value could be seen for higher concentrations of CA-NPs, it was not statistically significant.

Table 2 also shows the MI results of *Allium cepa* cells submitted to CA-NPs. CA-NPs caused a reduction in cell division in *Allium cepa* meristem cells at higher concentrations. The concentration of 100 µg·m$L^{-1}$ induced a statistically significant reduction of about 45% in MI when compared to the control group. The decrease in the MI index can be explained by the variations in the duration of the mitotic cycle due to the increase in the duration of the S phase [28], or it may be related to the blockage in the G1 stage, which suppresses DNA synthesis [29].

The genotoxic effects of CA-NPs on *Allium cepa* cells were also evaluated; however, no statistically significant changes were observed compared to the negative control. In addition, Table 2 shows that the micronucleus index (MNI) also showed no statistical difference compared to the control group. These results suggest that CA-NPs did not generate cytotoxic and genotoxic changes in *Allium cepa* meristematic cells. These results are of great interest, since these polymeric NPs can be used to deliver pesticides and/or fertilizers and offer many advantages, due to their highly stable nature and ability to encapsulate different active ingredients [30,31]. Thus, the question of their nanotoxicity is very important [32].

In general, bio-based polymeric nanoparticles are non-toxic and biodegradable and are considered superior nanocarriers [30]. Some polymers such as alginate, chitosan, poly(vinyl alcohol), poly(vinyl pyrrolidone), polyvinylidene fluoride, tripolyphosphate (TPP), and poly (ethylene glycol) methyl ether-block-lactide-co-glycolide (mPEG-PLGA) are commonly used in different application areas [33,34,35]. Currently, many polymeric nanoparticles are widely used in plant science to improve crop yields, but they cause some harm to the environment [36,37].

The present study provides more information about the environmental toxicity of CA-NPs, showing that this nanomaterial shows great potential as a nanocarrier, since we did not detect toxicity in the *Allium cepa* test, mainly for concentrations lower than 100 µg·m$L^{-1}$. However, the optimal doses of polymeric NPs for plant applications must be focused on, as there is a huge research gap in this area of plants [30].

## 4. Conclusions

The production of spherical CA-NPs with a main diameter of around 200 nm was possible through a modified emulsification solvent evaporation technique *o*/*w* (oil/water). FTIR analysis showed no great changes in the molecular vibrational modes of the cellulose acetate nanoparticle due to the synthesis process. The CA-NPs showed no changes in the root elongation and germination or in the mitotic, chromosomal aberration, and micronucleus indices, so they did not induce phytotoxic effects and did not generate cytotoxic and genotoxic changes in the *Allium cepa* test. CA-NPs show great potential for being used as nanocarriers to deliver pesticides and/or fertilizers; compared to other polymers, cellulose acetate is a low-cost and easier-to-find/produce biopolymer with a non-phytotoxic effect.

## Figures and Tables

**Figure 1 polymers-14-05022-f001:**
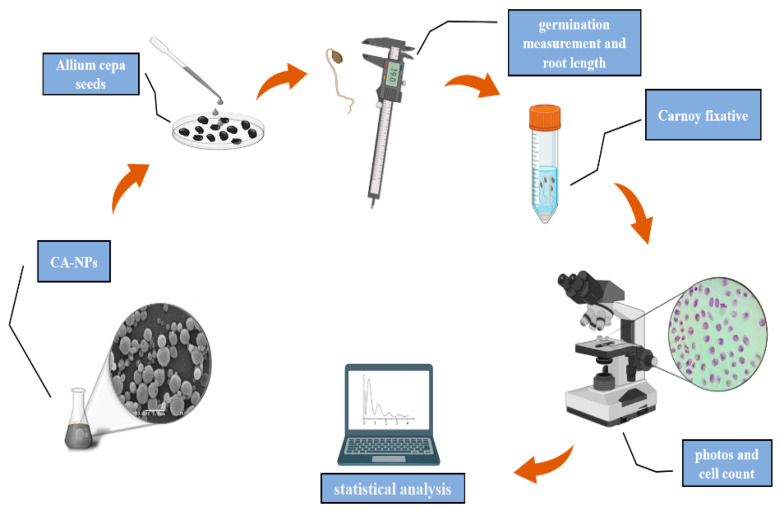
Schematic diagram of the *Allium cepa* assay.

**Figure 2 polymers-14-05022-f002:**
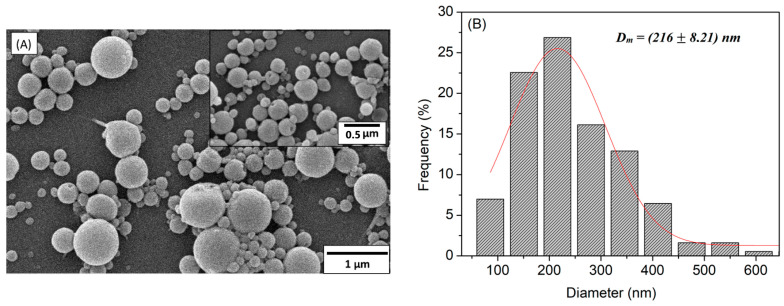
(**A**) Scanning electron micrographs of the cellulose acetate nanoparticles (CA-NPs) with an increase of ×20,000 and zoom of ×50,000; (**B**) histogram with a large distribution from 100 to 600 nm, with average diameter values of around 216 ± 8.21 nm.

**Figure 3 polymers-14-05022-f003:**
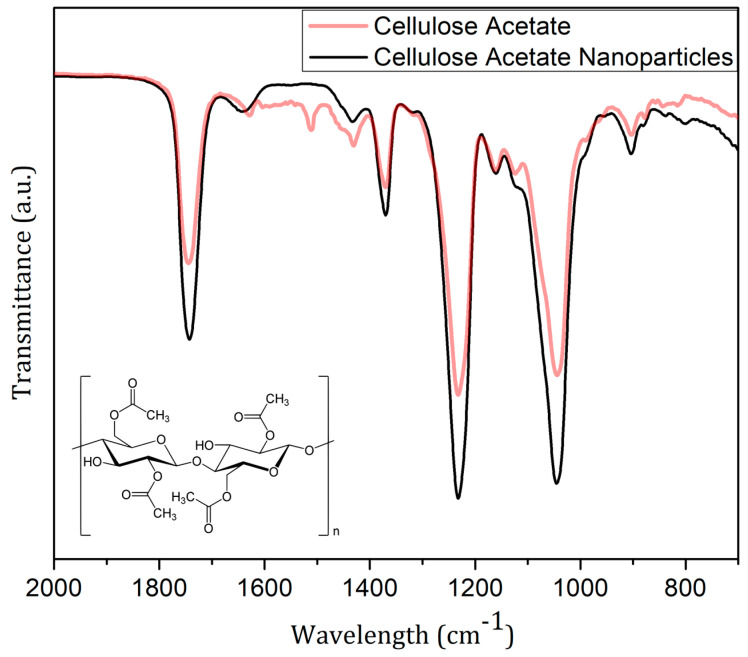
FTIR spectra of cellulose acetate nanoparticles (CA-NPs) (black line); cellulose acetate from Sigma-Aldrich (red line); and cellulose acetate chemical structure in detail.

**Table 1 polymers-14-05022-t001:** Hydrodynamic diameter (D_Size-DLS_), the Z average is the intensity-weighted mean hydrodynamic size (D_Z-Average-DLS_), and polydispersity index (PDI) of the CA-NPs. The nanoparticle diameter determined by scanning electron micrographs (D_SEM_) is also shown.

Particles	D_SEM_ (nm)	PDI	D_Size_-_DLS_ (nm)	D_Z-Average-DLS_ (nm)	Zeta (mV)
CA-NPs	216 ± 8.21	0.933	508.3 ± 49.7	1945 ± 87.6	−2.52

**Table 2 polymers-14-05022-t002:** Results of *Allium cepa* assay treated with solutions of CA-NPs. Data represent the mean ± standard error.

Treatment	MI (%)	CAI (%)	MNI (%)	IG (%)	REI (mm)
control (DI Water)	1.64 ± 0.38	0.08 ± 0.04	0.04 ± 0.04	70.00 ± 16.67	9.51 ± 0.88
12.5 µg·m$L^{-1}$	1.38 ± 0.44	0.12 ± 0.07	0.04 ± 0.02	74.44 ± 10.11	9.45 ± 0.46
25 µg·m$L^{-1}$	1.36 ± 0.64	0.08 ± 0.06	0 ± 0.00	65.56 ± 7.70	9.12 ± 2.30
50 µg·m$L^{-1}$	0.90 ± 0.22	0.04 ± 0.02	0.02 ± 0.02	65.56 ± 10.18	11.72 ± 2.15
100 µg·m$L^{-1}$	0.74 ± 0.28 *	0 ± 0.00	0.10 ± 0.06	67.78 ± 10.18	11.39 ± 1.51

* (asterisk) represents the significant difference between control and NP-treated samples (* *p* < 0.05); DI—Water Deionized; MI—mitotic index; CAI—chromosomal aberration index; MNI—micronuclei index; IG—germination index; REI—root elongation index.

## Data Availability

Data available on request.

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
