# Peer review of "Preparation and Phytotoxicity Evaluation of Cellulose Acetate Nanoparticles"

_polymers, 2022, doi:10.3390/polym14225022_

Round 1
Reviewer 1 Report
Manuscript Number: Polymers-1974411
The manuscript by Lima et al. titled “Preparation and phytotoxicity evaluation of cellulose acetate 2 nanoparticles” evaluated the phytotoxicity of cellulose acetate nanoparticles with a potential to use as a delivery vehicle in agriculture. Although cellulose acetate is a biobased material, it is essential to study its toxicity to plants. I want to suggest the following comments and clarify a few concerns which will further improve the article and make it suitable for Polymers.
1. The abstract should be improved by being concise and reflecting the key points of what has been done and what was the key findings.
2. The toxicity of cellulose acetate sheets to plants was first evaluated in 1952 by Karl Maramorosch in his published article in Science journal. This work might be valuable to cite.
3. Introduction, line 39-41, “Cellulose acetate hydrogel derived and ethylenediaminetetraacetic dianhydride (EDTAD)…”. Unclear sentence.
4. Sec 2.1, line 67-68, “Dichloromethane was used for the organic solution was used from Synth.” Please rephrase the sentence. Sec 2.2, defines RCF.
5. Sec 2.6, line 108-110, “Then the seeds were placed in a germination and growth chamber, type B.O.D (Biochemical Oxygen Demand), with temperature control of 25°C, a humidity of 85±5%, and a photoperiod of 12 in 12 h for 96 h”, is the word type necessary and little confused with photoperiod of 12 in 12 h for 96 h.
6. If possible, please provide a figure to depict the allium Cepa assay test.
7. Sec. 3, lines 132-133, please change the word high occurrence diameter to mean diameter.
8. Sec. 3, Line 140, is the alphabet e necessary after Figure 1(A).
9. Sec. 3, line 143, please change the word carboxylic to the carboxylic group.
10. For some numbers “,” is used, and for some “.” Is used. Its a little confusing.
11. The RAI (mm) should be REI (mm).
12. Line 189, please close the bracket for poly(vinylalcohol.
13. Use italics for all Allium cepa if appropriate.
14. Check for grammatical mistakes.
Author Response
- The abstract should be improved by being concise and reflecting the key points of what has been done and what was the key findings.
Thank you for the suggestion, we made proper changes in the abstract.
- The toxicity of cellulose acetate sheets to plants was first evaluated in 1952 by Karl Maramorosch in his published article in Science journal. This work might be valuable to cite.
Thank you for the suggestion, we made proper changes in the text.
- Introduction, line 39-41, “Cellulose acetate hydrogel derived and ethylenediaminetetraacetic dianhydride (EDTAD)…”. Unclear sentence.
Thank you for the suggestion, we made proper changes in the text.
- Sec 2.1, line 67-68, “Dichloromethane was used for the organic solution was used from Synth.” Please rephrase the sentence. Sec 2.2, defines RCF.
Thank you for the suggestion, we made proper changes in the text.
- Sec 2.6, line 108-110, “Then the seeds were placed in a germination and growth chamber, type B.O.D (Biochemical Oxygen Demand), with temperature control of 25°C, a humidity of 85±5%, and a photoperiod of 12 in 12 h for 96 h”, is the word type necessary and little confused with photoperiod of 12 in 12 h for 96 h.
Thank you for the suggestion, we made proper changes in the text.
- If possible, please provide a figure to depict the allium Cepa assay test.
Thank you for the suggestion, we introduced the Figure 1 in the text.
- Sec. 3, lines 132-133, please change the word high occurrence diameter to mean diameter.
Thank you for the suggestion, we made proper changes in the text.
- Sec. 3, Line 140, is the alphabet e necessary after Figure 1(A).
Thank you for the suggestion, we made proper changes in the text.
- Sec. 3, line 143, please change the word carboxylic to the carboxylic group.
Thank you for the suggestion, we made proper changes in the text.
- For some numbers “,” is used, and for some “.” Is used. Its a little confusing.
Thank you for the suggestion, we made proper changes in the text.
- The RAI (mm) should be REI (mm).
Thank you for the suggestion, we made proper changes in the text.
- Line 189, please close the bracket for poly(vinylalcohol.
Thank you for the suggestion, we made proper changes in the text.
- Use italics for all Allium cepa if appropriate.
Thank you for the suggestion, we made proper changes in the text.
Reviewer 2 Report
Authors should revise the manuscript as follows:
1. The author should add an illustration for the theme of the manuscript that can tell the whole story at a glance to the readers.
2. Author should refer to the following very important article which has described biodegradabile polymer designing application:
A.Effect of polyethene glycol on properties and drug encapsulation–release performance of biodegradable/cytocompatible agarose–polyethene glycol–polycaprolactone amphiphilic co-network gels." ACS applied materials & interfaces 8.5 (2016): 3182-3192.
B. Liquid prepolymer-based in situ formation of degradable poly (ethylene glycol)-linked-poly (caprolactone)-linked-poly (2-dimethylaminoethyl) methacrylate amphiphilic conetwork gels showing polarity driven gelation and bioadhesion." ACS Applied Bio Materials 1.5 (2018): 1606-1619.
C.Recent advances in the potential applications of luminescence-based, SPR-based, and carbon-based biosensors." Applied Microbiology and Biotechnology (2022): 1-27.
3. Author should explain why Cellulose chooses not other polysaccharides are there any particular advantages to using cellulose?
4. Author should add all images of the different experimental processes of the Allium cepa assay
5. AUthor should add the characterization of NPs and stability study data.
The author can follow the following very relevant research article synthesis and multi‐responsive self‐assembly of cationic poly (caprolactone)–poly (ethylene glycol) multiblock copolymers." Chemistry–A European Journal 23.34 (2017): 8166-8170.
Author Response
- The author should add an illustration for the theme of the manuscript that can tell the whole story at a glance to the readers.
Thank you for the suggestion, we introduced the Figure 1 in the text.
Author should refer to the following very important article which has described biodegradabile polymer designing application:
A.Effect of polyethene glycol on properties and drug encapsulation–release performance of biodegradable/cytocompatible agarose–polyethene glycol–polycaprolactone amphiphilic co-network gels." ACS applied materials & interfaces 8.5 (2016): 3182-3192.
B. Liquid prepolymer-based in situ formation of degradable poly (ethylene glycol)-linked-poly (caprolactone)-linked-poly (2-dimethylaminoethyl) methacrylate amphiphilic conetwork gels showing polarity driven gelation and bioadhesion." ACS Applied Bio Materials 1.5 (2018): 1606-1619.
C.Recent advances in the potential applications of luminescence-based, SPR-based, and carbon-based biosensors." Applied Microbiology and Biotechnology (2022): 1-27.
Thank you for the suggestion
- Author should explain why Cellulose chooses not other polysaccharides are there any particular advantages to using cellulose?
Thank you for the suggestion, the choose for CA was only based in the great offer of this material, and the eventual high demand if applicable for agricultural proposes.
- Author should add all images of the different experimental processes of the Allium cepa assay
Thank you for the suggestion, but we believe it is not necessary, usually this type of study does not demand these images which only add excess information without scientific and/or analytical value, for example these published works do not bring images of the entire process.
Guilger, M., Pasquoto-Stigliani, T., Bilesky-Jose, N. et al. Biogenic silver nanoparticles based on trichoderma harzianum: synthesis, characterization, toxicity evaluation and biological activity. Sci Rep 7, 44421 (2017). https://doi.org/10.1038/srep44421
Casillas-Figueroa F, Arellano-García ME, Leyva-Aguilera C, Ruíz-Ruíz B, Luna Vázquez-Gómez R, Radilla-Chávez P, Chávez-Santoscoy RA, Pestryakov A, Toledano-Magaña Y, García-Ramos JC, Bogdanchikova N. Argovit™ Silver Nanoparticles Effects on Allium cepa: Plant Growth Promotion without Cyto Genotoxic Damage. Nanomaterials. 2020; 10(7):1386. https://doi.org/10.3390/nano10071386
- AUthor should add the characterization of NPs and stability study data.
The author can follow the following very relevant research article synthesis and multi‐responsive self‐assembly of cationic poly (caprolactone)–poly (ethylene glycol) multiblock copolymers." Chemistry–A European Journal 23.34 (2017): 8166-8170.
Thank you for the suggestion
Reviewer 3 Report
The current work has merit but is very brief in the results and discussion. It is more a communication, not a full length research article. Therefore I suggest to authors to make some changes:
The introduction should be enriched with a figures displaying the key chemical structures. Also much more framing is needed. What is new compared to the field?
The Results and Discussion needs more intro. It is too direct. Also highlight more the outlook.
To be clear the body is fine but it needs more intro and outlook to enable full appreciation.
Small comments.
Mw needs to be explained for a general reader; also units are missing of g/mol
RCF is not fully explained
Author Response
The introduction should be enriched with a figure displaying the key chemical structures. Also much more framing is needed. What is new compared to the field?
The Results and Discussion needs more intro. It is too direct. Also highlight more the outlook.
To be clear the body is fine but it needs more intro and outlook to enable full appreciation.
Small comments.
Mw needs to be explained for a general reader; also, units are missing of g/mol
RCF is not fully explained
Thank you for the suggestion, we tried to improve the manuscript content by adding a Figure scheme of the experiment and other information in the text.
Round 2
Reviewer 2 Report
NA
Author Response
There is no comments to be answered